# Development of a Radioactive Sorting and Volume Reduction System for Radioactive Contaminated Sandy Soil Using Plastic Scintillator and NaI Detectors

**DOI:** 10.3390/s25247458

**Published:** 2025-12-08

**Authors:** Chengzhou Fan, Zhenxing Liu, Jinshuai Yang, Rui Li, Jianbo Yang

**Affiliations:** 1Northwest Institute of Nuclear Technology, Xi’an 710065, China; 2School of Physics and Electronic Engineering, Sichuan University of Science & Engineering, Zigong 643002, China

**Keywords:** soil sorting, volume reduction, plastic scintillator detector, NaI detector

## Abstract

Radioactively contaminated sandy soil is commonly encountered during nuclear facility decommissioning and nuclear accident response, and its rapid sorting and volume reduction are crucial for achieving waste minimization and lowering remediation costs. This study designed and developed a radiation measurement system based on a large-volume plastic scintillator and a NaI array detector, focusing on the design, implementation, and performance validation of its radiation detection and signal processing subsystems. The system employed differential measurement to obtain the net radioactive count rate of sandy soil, while enhancing energy spectrum stability through programmable gain control and temperature stabilization. Experimental results demonstrated that both plastic scintillator arrays effectively achieved dynamic background subtraction within a 1.8 s measurement cycle, with net count rate errors controlled below 10%. The NaI detector array achieved an energy resolution better than 8% at 662 keV, with the peak channel drift within ±1 channel. Rapid activity measurements for radioactive sources such as ^241^Am and ^137^Cs exhibited errors below 10%, meeting the key technical requirements for sandy soil separation and volume reduction. These findings provided data support and methodological reference for subsequent system integration and engineering application of sorting and volume reduction equipment.

## 1. Introduction

Radioactively contaminated sandy soil is widely present in decommissioned nuclear facilities and nuclear accident cleanup scenarios [1,2,3,4,5]. Its rapid and effective sorting and volume reduction are crucial for ensuring public safety and minimizing disposal costs [6,7,8]. Various detection technologies have been developed for radiation mapping and contamination assessment in post-accident scenarios [9,10,11]. Employing radiometric measurement techniques, contaminated sandy soil can be sorted based on activity levels into two groups: those exceeding standards and those eligible for exemption. Alternatively, it can be classified into three categories: highly radioactive (requiring engineered disposal), moderately radioactive (suitable for controlled landfill), and low-level radioactive or exempt (eligible for unrestricted release). This approach enables the in situ backfilling or resource utilization of large volumes of low-level or exempt sandy soil, representing a key technological pathway for achieving waste minimization and reducing remediation costs [12].

In this context, radioactive measurement-based sorting and volume reduction technologies have attracted increasing attention [13]. Commercially available systems such as the Segmented Gate System (SGS) have proven effective in site remediation applications [14,15]. However, their core technologies remain proprietary, and technical specifications are not accessible through standard procurement processes. Consequently, the development of application-specific sorting and volume reduction systems is critical for facilitating decommissioning and remediation operations.

This study aims to develop a radioactive measurement system based on large-volume plastic scintillator detectors and arrayed NaI detectors, with a focus on theoretical design and experimental validation of its electronics system. By verifying the performance of the detection modules, this work seeks to evaluate the system’s capability to meet the key performance requirements for waste sorting and volume reduction equipment, thereby providing data support and methodological references for subsequent equipment integration and demonstration applications.

## 2. Materials and Methods

### 2.1. System Design

#### 2.1.1. Overall System Architecture

The radioactive measurement system for the sorting and volume reduction device was designed to enable rapid and accurate measurement of radioactive substances in the tested sandy soil, providing measurement data for radiation-based classification and sorting of the sandy soil. As shown in Figure 1, this device consists of two main components: the conveying module and the detection module. The conveying module employs a conveyor belt structure to ensure continuous and stable transport of the sandy soil samples.

The detection module was designed to meet stringent operational requirements for high-throughput sorting applications. The system specification mandates a minimum processing capacity of 30 t/h with a sorting precision of ≤15 kg per batch (assuming a sandy soil density of 1.6 t/m^3^, equivalent to approximately 18 m^3^/h). These parameters dictate a measurement cycle of 1.8 s, derived as follows: at 30 t/h throughput with 15 kg batches, the system must process 2000 discrete samples per hour (30,000 kg ÷ 15 kg/sample), corresponding to a temporal resolution of 1.8 s per measurement (3600 s ÷ 2000 samples). This operational constraint imposes significant challenges on detector performance, as conventional measurement systems exhibit substantial statistical fluctuations in results obtained within such abbreviated timeframes. Therefore, we adopted a strategy combining large crystals with rapid signal acquisition and processing, enhancing system detection efficiency by increasing crystal size [16]. Specifically, the detector system is mounted above the conveyor belt in a sequential arrangement. As the conveyor belt advances, each section first passes beneath Plastic Scintillator #1, which measures the environmental background radiation when no sample is present on the belt at that measurement position. Subsequently, as sandy soil is deposited onto the belt and transported forward, the same belt section—now carrying the sample—passes beneath Plastic Scintillator #2. At this position, Plastic Scintillator #2 measures the combined signal from the sandy soil sample, conveyor belt contribution, and environmental background. By subtracting the background measurement from Plastic Scintillator #1 (using a pre-calibrated correction coefficient k, as detailed in Section 2.3.2, Equation (1)) from the total signal measured by Plastic Scintillator #2, the net radioactivity of the sandy soil is determined in real time. This differential measurement approach effectively eliminates temporal variations in environmental background that would otherwise introduce systematic errors in short-duration measurements.

When the calculated net count rate exceeds a predefined threshold indicating potential contamination, the NaI array measurement system is automatically triggered for nuclide identification and activity quantification. The array comprises four large-volume NaI detectors (NaI #1–#4, 100 mm × 100 mm × 400 mm each, as specified in Section 2.1.3) arranged side-by-side to cover the full width of the conveyor belt cross-section. This configuration is optimized for low-to-moderate activity samples. For high-activity samples that would cause detector saturation or pulse pile-up in the array system, the measurement automatically switches to the independent NaI detector #5, which features a smaller crystal volume (Φ 50 × 50 mm) and is positioned to handle high count rates without signal congestion. After completing the radioactivity measurement sequence, the sample proceeds to downstream sorting mechanisms that execute diversion actions based on the sorting results.

#### 2.1.2. Plastic Scintillator Detector Design

The operating environment of the plastic scintillator detector in this study could involve significant temperature fluctuations, mechanical vibrations, and even impacts. To ensure stable detector operation, the detector housing employed a stainless steel frame with aluminum alloy side panels. All six internal surfaces were equipped with thermal insulation layers and sealing gaskets. Vibration dampers were installed at the base to mitigate mechanical impacts during transportation. Additionally, a constant-temperature gas channel was incorporated to maintain a stable operating environment for the plastic scintillator. The plastic scintillators used in the radiation measurement system are polystyrene-based, specifically the UPS-923A model (Amcrys, Kharkov, Ukraine), each with dimensions of 800 mm × 400 mm × 50 mm, surrounded by solid white reflective material to enhance light collection efficiency. Two 2-inch photomultiplier tubes (Hamamatsu R6231 or equivalent, Hamamatsu Photonics, Hamamatsu, Japan) were coupled to opposite ends of the scintillator for signal readout. Each PMT was shielded with 0.6 mm thick mu-metal to minimize the influence of external magnetic fields. A cross-sectional view of the plastic scintillator detector housing is shown in Figure 2.

#### 2.1.3. NaI Array Detector Design

The energy resolution of plastic scintillators is limited, rendering their measurement results unsuitable for nuclide identification. Therefore, this study designed an array system comprising four large crystals (100 mm × 100 mm × 400 mm) and one small crystal (Φ 50 mm × 50 mm) of NaI detectors (Saint-Gobain Crystals, Hiram, OH, USA), covering a 40 cm width of the conveyor belt. The NaI detectors were also housed within stainless steel enclosures, employing the same shielding configuration as the plastic scintillators to ensure NaI crystal stability. A cross-sectional view of the NaI test chamber is shown in Figure 3.

#### 2.1.4. Temperature Control System

Considering that both the luminescence efficiency of NaI and the photomultiplier tube (PMT gain) are temperature-sensitive, and that the ambient temperature during field operation may range from −20 °C to +50 °C, the system incorporates a temperature stabilization control design to ensure the accuracy and stability of radiation measurements.

The selected cabinet air conditioner employs two independent, isolated circulation systems—one internal and one external—to regulate cabinet temperature through heat exchange, achieving efficient climate control with both heating and cooling capabilities. The conditioned air is conveyed via plastic corrugated tubing and stainless steel connectors to each of the three detector measurement chambers, where it recirculates back to the air conditioner through an intake port, creating a complete airflow loop. This ensures the detector crystals and electronic components operate at an optimal and constant temperature, effectively minimizing the impact of temperature fluctuations on nuclear radiation measurements.

To meet the special requirements of field radiation measurement, this solution incorporates a stainless steel protective cover for the air conditioner, ensuring secure installation during measurement and transportation while effectively shielding it from sand and dust erosion. The rear of the enclosure features an intake sleeve, exhaust housing, and thermal insulation foam. This integrated design maintains optimal temperature and humidity conditions within the cabinet while isolating external dust, thereby extending the lifespan of electrical components and enhancing system operational reliability (as shown in Figure 4).

### 2.2. Electronics System Design

To achieve rapid and continuous measurement of radioactivity in sandy soil, a digital energy spectrum measurement system was designed specifically optimized to handle the high count rates and large signal dynamic range characteristic of large-volume scintillator detectors. Adopting a modular design philosophy, the system comprised a detector module, a signal conditioning and amplification module, a data acquisition and processing module, and a power management module. The overall system architecture is shown in Figure 5, with each module designed to achieve the core objectives of high energy resolution and high stability.

#### 2.2.1. Signal Conditioning and Amplification Module

The faint current signal output by the detector via the photomultiplier tube is first converted into a voltage signal and preliminarily amplified by a low-noise preamplifier. Subsequently, the signal enters the conditioning and amplification board, sequentially passing through a pole-zero cancellation circuit (Figure 6) and a programmable gain amplifier circuit (Figure 7). The former effectively eliminates pulse trailing edge overshoot through frequency-domain zero compensation, shaping broad pulses into narrow ones while suppressing baseline drift and pulse stacking. The latter, composed of a digital-to-analog converter (DAC) and operational amplifier, dynamically adjusts signal amplitude under varying measurement conditions to ensure it remains within the optimal sampling range of the analog-to-digital converter (ADC). The introduction of programmable gain not only enables the system to adapt to signal amplitude variations across different radioactivity levels but also maintains spectral stability through automatic compensation when peak channel positions drift due to temperature changes. This design ensures the system maintains long-term stable operation even in complex environments.

#### 2.2.2. Data Acquisition and Processing Module

The ADC serves as the core component of a digital energy spectrum system, converting analog pulses into digital signals with high precision for real-time processing. Its performance directly determines the system’s energy resolution. This study employed a high-speed 12-bit ADC chip with a maximum sampling rate of 20 MHz. The FPGA provided precise sampling clocks and controlled the sampling process, as illustrated in Figure 8. The ADC was configured in differential input mode with an input voltage range of 0–2 V, outputting data in offset binary format.

The selected FPGA, a Xilinx XC3S500E (Xilinx, San Jose, CA, USA), employed a sliding window minimum-average algorithm for real-time baseline estimation. FPGA-based sensor systems have been widely recognized for their reconfigurability and low power consumption in complex signal processing applications [17]. A trapezoidal shaping algorithm with adjustable parameters enhanced the pulse signal-to-noise ratio and performed preliminary separation of overlapping pulses. The system identified pile-up events by detecting consecutive rising edges following the falling edges of preceding pulses. Processed energy spectrum data and status information were transmitted to the main controller (STM32F103VET6 MCU, STMicroelectronics, Geneva, Switzerland) via a parallel bus. The MCU handled system control, data integration, and baseline drift detection, communicating with the host computer via SPI and serial interfaces to display spectra and upload data. The peripheral circuit diagram of the MCU is shown in Figure 9.

#### 2.2.3. Power Supply System

The system employed independent and isolated analog and digital power supplies. A switching power supply module converted the 220 V AC input into +12 V and +5 V DC outputs, which voltage regulator circuits then regulated to produce the required operating voltages. For analog circuits including preamplifiers and signal conditioning modules, the design utilized low-noise linear regulators (LDOs) ADP7142/ADP7182 (Analog Devices Inc., Wilmington, DE, USA) to generate precise ±5 V symmetrical power supplies, as shown in Figure 10.

The low-voltage digital power supply circuit is shown in Figure 11. For digital circuits including the FPGA, MCU, and ADC, the design selected the high-efficiency switching regulator IC ADP5135 (Analog Devices Inc., Wilmington, DE, USA). By configuring feedback resistors, this IC generated the required 1.2 V, 2.5 V, and 3.3 V power supplies. The three outputs were enabled sequentially to meet strict power-up sequencing requirements. Throughout the power supply design, the analog and digital grounds were connected at a single point at the power input, and extensive decoupling capacitors were placed to maximize suppression of digital noise interference with analog signals.

### 2.3. Experimental Setup

#### 2.3.1. Radioactive Sources

To verify the performance and feasibility of the radiation dosimetry system, a controlled experimental measurement environment was established using calibrated gamma radiation sources. Three sealed sources, manufactured by Beijing Atom High-Tech Co., Ltd. (Beijing, China), were employed in the validation tests: Americium-241 (^241^Am), Cesium-137 (^137^Cs), and Cobalt-60 (^60^Co).

The specific parameters of these sources are shown in Table 1. It should be noted that the ^241^Am radiation source is a surface source with a specified emission rate for alpha particles of 1.95 × 10^+5^/2π·min. Therefore, the activity of the ^241^Am source described herein can be considered to be 3250 Bq.

#### 2.3.2. Experimental Procedures

To validate the performance of the radiation measurement system, a series of characterization tests were conducted using the aforementioned three gamma radiation sources.

Background subtraction coefficient calibration: In passive mode, both plastic scintillators were placed on a standard ceramic tile floor, outputting data every 1.8 s to measure the count rates of plastic scintillators #1 and plastic scintillators #2. Assuming a proportional relationship exists between the two detectors under identical environmental conditions, the background correction coefficient *k* was calculated using(1)$$C_{\#1}=k\cdot C_{\#2},$$
where $C_{\#1}$ and $C_{\#2}$ represent the average background count rates of plastic scintillators #1 and plastic scintillators #2, respectively.

Subsequently, active validation tests were performed by sequentially positioning the three radioactive sources at the geometric center between the two plastic scintillators, 3 cm above the detector base, with measurements taken at 1.8 s, 3.6 s, and 60 s intervals. Background correction was applied using Equation (1) to obtain net count rates.

NaI detector energy calibration: Energy spectrum stability was verified using ^241^Am and ^60^Co in the range of 50 keV to 1 MeV, following the linear relationship(2)E=aD+b,
where *E* represents energy, *D* denotes the channel address, and *a* and *b* are calibration coefficients. Following calibration, energy resolution was assessed using a ^137^Cs source.

Spectral stability assessment: A ^137^Cs source was positioned at the geometric center of the detector array with measurements performed over 1.8 s cycles. A total of 70 repeated measurements were conducted to evaluate the stability of peak positions and channel addresses.

Activity measurement validation: To evaluate the system’s activity measurement performance under operational time constraints, standard ^241^Am and ^137^Cs sources were tested. For NaI array detectors (#1–#4), sources were placed at the geometric center of the detector; for NaI #5, sources were positioned directly beneath the detector. All measurements were performed with 1.8 s acquisition times to simulate sorting operation conditions.

## 3. Results

### 3.1. Plastic Scintillator Performance

The primary function of the plastic scintillation detector is the rapid measurement of radioactive materials and triggering of the NaI detector. To ensure fast and accurate measurement of the net radioactivity count rate in sandy soil, real-time background subtraction must be applied to the measurement results from both plastic scintillation detectors. Therefore, the background subtraction coefficients for both detectors are first calibrated. The detector counts obtained from the experimental measurements are shown in Figure 12.

Substituting the experimental data into Equation (1), the background subtraction coefficient was calculated as k = 1.2031. To validate the derived correction coefficients, active measurements were conducted. The net count rates for the same source measured at different durations (1.8 s, 3.6 s, and 60 s) showed excellent consistency across both detector sets, as shown in Figure 13. Error bars indicate the statistical uncertainty calculated based on Poisson statistics as(3)$$\sigma_{R_{net}}=\frac{\sqrt{N_{total}+N_{bg}}}{t}\;,$$
where $N_{total}$ and $N_{bg}$ are the total counts and background counts, respectively, and *t* is the measurement live time. The corresponding fractional (relative) uncertainty of the net count rate is(4)$$\frac{\sigma_{R_{net}}}{R_{net}}=\frac{\sqrt{N_{total}+N_{bg}}}{N_{total}-N_{bg}}\;$$
and is expressed as a percentage by multiplying the left-hand side by 100%. Using this expression, the statistical uncertainties were approximately 0.55–0.56% for the 1.8 s measurements, 0.39–0.41% for the 3.6 s measurements, and 0.09–0.10% for the 60 s measurements.

The count rate error for the same source was controlled within 9%, confirming the correctness and reliability of the dynamic background deduction model. The experiment demonstrated that the two large-volume plastic scintillator systems in this study can effectively measure the radioactivity of a radioactive source within 1.8 s.

### 3.2. NaI Detector Performance

The NaI array detector is tasked with identifying radionuclides and measuring their activity. We conducted experiments according to the system design requirements to verify the spectral stability of the system. The calculated calibration parameters for the five NaI detectors are shown in Table 2.

Following the energy calibration, the NaI detectors underwent energy spectrum testing. The detectors were calibrated using ^137^Cs, with the measured experimental spectrum shown in Figure 14. Results indicate that all four detectors in the array exhibit energy resolution below 8% at 662 keV. The standalone NaI detector #5 demonstrates approximately 9% resolution at the same energy, meeting design specifications.

The stability of the detector’s output energy spectrum directly impacts the accuracy of nuclide identification. To verify the spectral stability of the four large crystals, a ^137^Cs source was placed at the center of the detector array. Measurements were taken over a 1.8 s cycle, with 70 repeated measurements as shown in Figure 15. The test demonstrates that within the 1.8 s measurement cycle, fluctuations in peak position and channel address were consistently controlled within ±1 channel.

### 3.3. System Sorting Capability Validation

To validate the activity measurement performance of the radiation measurement system under short time windows, tests were conducted using standard ^241^Am and ^137^Cs sources. Experimental results obtained within a 1.8 s measurement period are shown in Figure 16. The results indicate that the average measurement errors for the ^241^Am and ^137^Cs activities detected by the array NaI detectors #1–#4 were 6% and 3%, respectively. For the NaI detector #5, the measurement errors were 11% and 9%, respectively. These results demonstrate that all detectors can meet the accuracy requirements for activity discrimination in rapid sorting, thereby providing reliable decision-making support for subsequent sorting execution mechanisms.

## 4. Discussion

Compared to the SGS radioactive soil sorting device, which has been deployed in multiple international remediation projects with documented operational results, the radioactive sandy soil sorting and volume reduction measurement system proposed in this study exhibit fundamental differences in structural layout, detection strategy, and electronics architecture. Developed by Sandia National Laboratories in the late 1990s, the SGS system typically operates by monitoring total counts as material passes through detector arrays on a conveyor belt. Material exceeding preset thresholds is diverted and recovered via mechanical gates, achieving soil volume reduction in several demonstration projects. A single pass can analyze two nuclides, with a theoretical processing capacity of approximately 400 t/h and a spectrum acquisition rate of about 0.5 s per pass. At this rate, the minimum sorting precision is approximately 50 kg [15]. Public project documentation also indicates that this sorting method has been deployed multiple times in practice. However, its specific operating parameters—including conveyor speed, detection limits, and throughput rates—vary with site-specific configurations, and standardized performance metrics are not consistently reported in publicly available documentation [14].

This study prioritizes precision in net counting and spectral stability during short measurement cycles in its overall structure and electronics design. The differential measurement method is the core technology enabling high-precision, rapid measurements. Due to significant variations in ambient background radiation levels over time and location [18,19], traditional single-channel measurements struggle to achieve stable background subtraction within short timeframes. In a conventional approach where background is measured separately from sample measurements, such temporal variations introduce systematic errors that cannot be distinguished from true sample activity changes.

The sequential differential measurement approach implemented in this study addresses this limitation through spatially separated but temporally proximate measurements. As shown in Figure 1, Plastic Scintillator #1 measures the bare conveyor belt section immediately before Plastic Scintillator #2 measures the same belt section carrying the sandy soil sample. The temporal separation between these two measurements is determined by the belt speed and detector spacing, typically on the order of seconds—short enough that environmental background remains essentially constant between the two readings. This near-simultaneous measurement strategy ensures that both detectors experience nearly identical environmental conditions, making the differential calculation robust against background fluctuations.

As demonstrated in Section 3.1, the dual-detector system maintained net count rate consistency across measurement durations ranging from 1.8 s to 60 s, with relative deviations below 9% for all tested sources (Figure 13). Notably, the background subtraction coefficient k = 1.2031 (derived from geometric efficiency differences between the two detector positions) remained stable throughout extended testing, validating the assumption of proportional background response. This performance indicates that the sequential differential approach effectively suppresses environmental variations that would otherwise dominate the measurement uncertainty budget in short-duration measurements. Consequently, the system achieved reliable discrimination capability even when the sample activity was only 2–3 times above the instantaneous background level—a challenging scenario where conventional methods would require substantially longer acquisition times or frequent background updates to achieve comparable statistical confidence [20].

It should be noted that the 1.8 s measurement window inherently introduces statistical uncertainties in count rate determination. During the background subtraction coefficient calibration process, statistical fluctuations in detector counts occasionally result in a negative net count rate—specifically when the reading from plastic scintillator #2, after correction by coefficient k, is slightly lower than that from plastic scintillator #1. This phenomenon is attributable to Poisson counting statistics rather than systematic measurement error. For the intended application of sandy soil sorting and volume reduction, operational thresholds are established with sufficient margins to accommodate such statistical variations. The observed ≤10% error in net count rate determination remains well within the acceptable tolerance for classification decisions, as the sorting criteria distinguish between materials requiring disposal (>exemption limits) and those eligible for unrestricted release or reuse.

Building upon this foundation, several key design elements were specifically optimized to address challenges such as short measurement durations, high statistical fluctuations, and variations in ambient temperature.

Under field conditions with a wide dynamic range of count rates, Programmable Gain Control (PGC) proved critical for maintaining spectral consistency and peak stability by ensuring signal amplitudes remained within the optimal range of the ADC, thereby avoiding both saturation and quantization noise issues [21]. The temperature stabilization system operated independently to ensure the long-term stability of the electronics and detectors. Both the light emission efficiency of plastic scintillator and NaI(Tl) crystals and the gain of photomultiplier tubes are temperature-sensitive; temperature fluctuations directly cause peak drift and variations in count rate [22,23,24,25]. The system’s thermostatic chamber and airflow channel design maintained controlled temperature differentials within the detector housing. These two mechanisms worked in parallel: temperature control minimized drift from thermal effects on scintillator light yield and PMT gain characteristics, while PGC dynamically adjusted gain based on real-time count rate variations to maintain optimal signal amplitudes. Experimental results showed that under the combined effects of temperature stabilization and PGC, peak position fluctuations in the NaI array were controlled within ±1 channel during 70 repeated ^137^Cs tests, validating the effectiveness of these integrated controls in maintaining short-term energy spectrum stability in complex environments.

This study employs two 800 mm × 400 mm × 50 mm plastic scintillators to achieve high detection efficiency for rapid net counting discrimination. Following threshold triggering, energy spectrum measurement and nuclide identification are performed by a 4 × (100 mm × 100 mm × 400 mm) NaI(Tl) crystal array. Experimental results demonstrated that the large-volume crystal design achieved a favorable balance between statistical efficiency and energy resolution.

In summary, the sorting measurement system developed in this study achieved stable discrimination of radioactive net counts and nuclide activity within a measurement cycle of 1.8 s. This capability was realized through the synergistic optimization of differential measurement, programmable gain control, constant temperature regulation, and large-volume detector design. Compared to traditional systems like SGS, which rely on high-throughput mechanical grading, this system is better suited for rapid on-site discrimination and quantitative measurement under dynamic background conditions, providing technical support for the domestic production and emergency application of radioactive contaminated sand and soil sorting and volume reduction equipment.

## 5. Conclusions

This study addresses the practical need for sorting and volume reduction of radioactive contaminated sandy soil by proposing and validating a radiation measurement system composed of a large-volume plastic scintillator and a NaI detector array. Research findings indicated that the plastic scintillator detector, combined with a background dynamic subtraction model, enabled rapid radioactivity measurement of sandy soil within short measurement cycles, with errors kept within an acceptable range for engineering applications. The NaI array detector demonstrated excellent energy resolution and stability, effectively supporting nuclide identification and activity measurement. The electronics system enhanced adaptability and long-term operational stability through programmable gain control and temperature stabilization. Experimental results validated the system’s feasibility for achieving net radiation counting and activity discrimination within short time windows, providing reliable data support for sorting operations. In summary, the proposed radiation measurement system lays the foundation for subsequent device integration, field application, and further optimization studies.

## Figures and Tables

**Figure 1 sensors-25-07458-f001:**
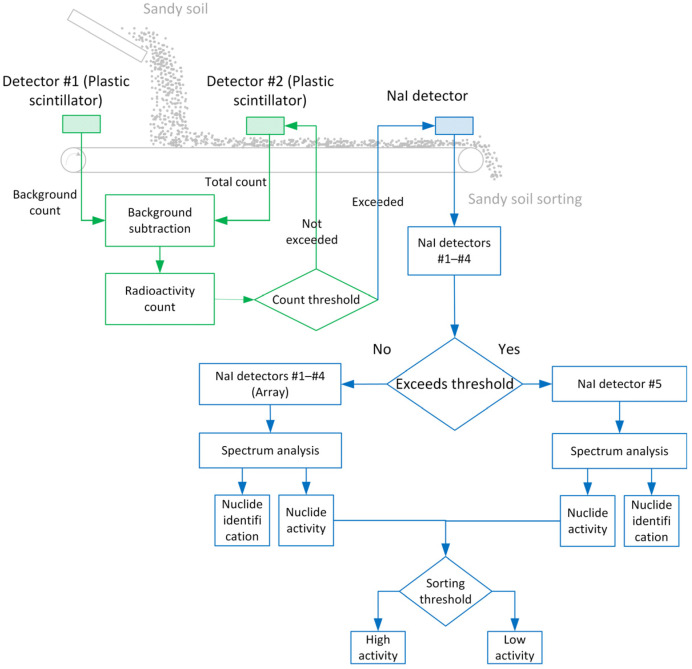
Workflow of the Radioactive Sorting and Volume Reduction System.

**Figure 2 sensors-25-07458-f002:**
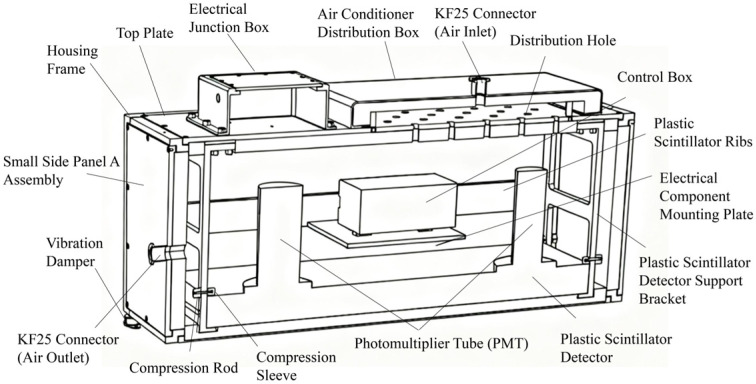
Cross-section of the plastic scintillator detector housing.

**Figure 3 sensors-25-07458-f003:**
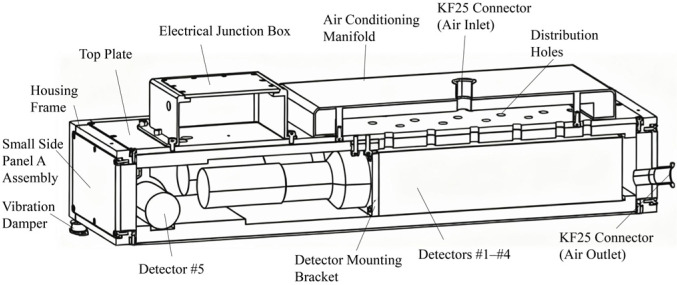
Cross-section of the NaI detector housing.

**Figure 4 sensors-25-07458-f004:**
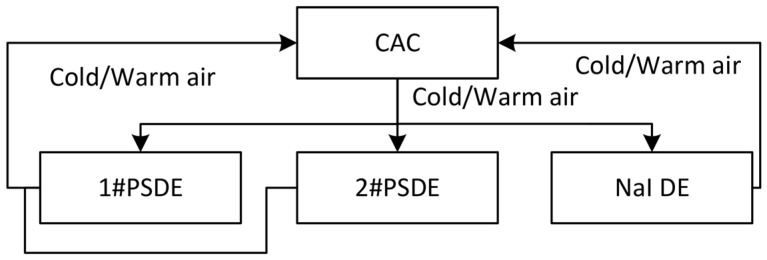
Schematic diagram of the temperature control system. Abbreviations: PSDE—Plastic Scintillator Detector Enclosure; NaI DE—NaI Detector Enclosure; CAC—Cabinet Air Conditioner.

**Figure 5 sensors-25-07458-f005:**
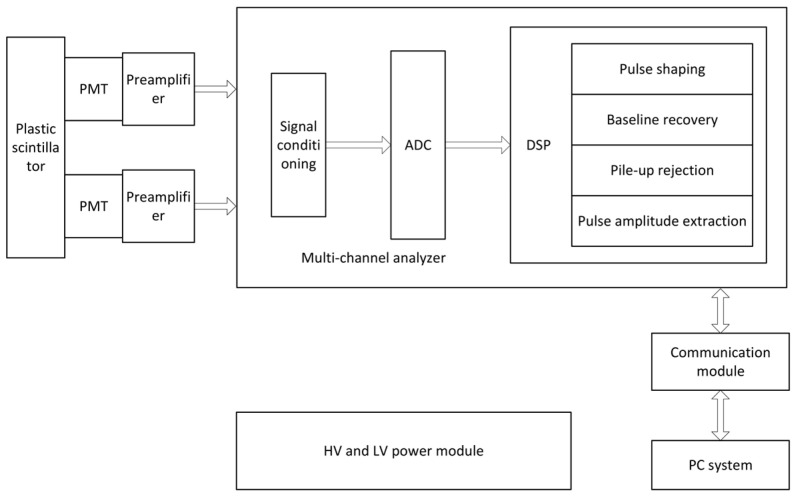
Block diagram of the digital energy spectrum measurement system.

**Figure 6 sensors-25-07458-f006:**
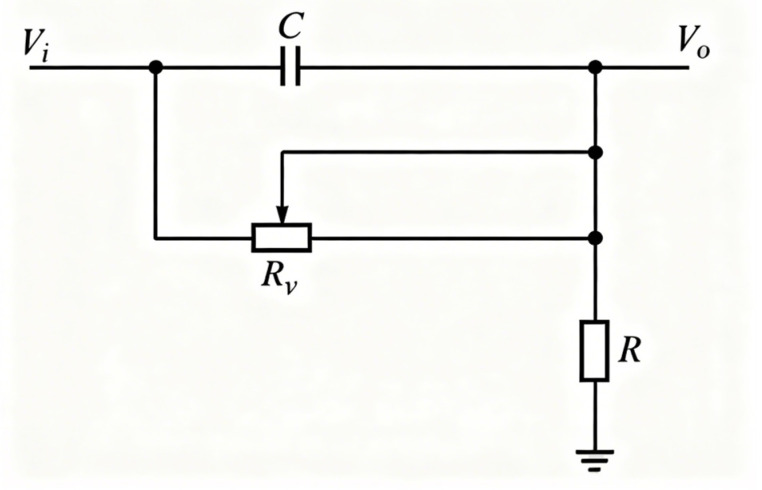
Schematic diagram of pole-zero cancellation circuit.

**Figure 7 sensors-25-07458-f007:**
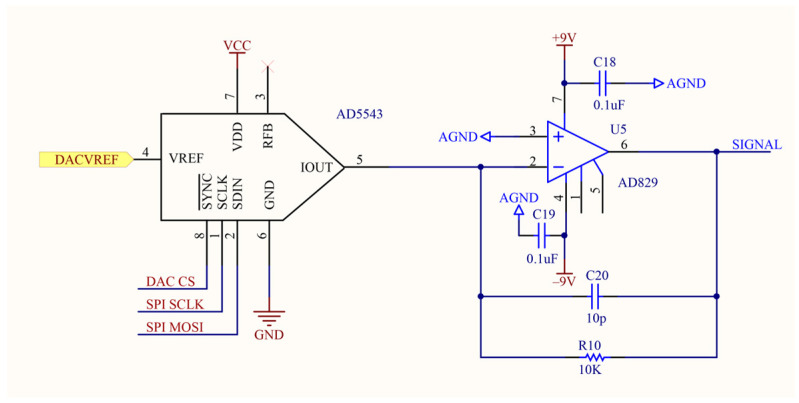
Schematic diagram of programmable gain amplifier circuit.

**Figure 8 sensors-25-07458-f008:**
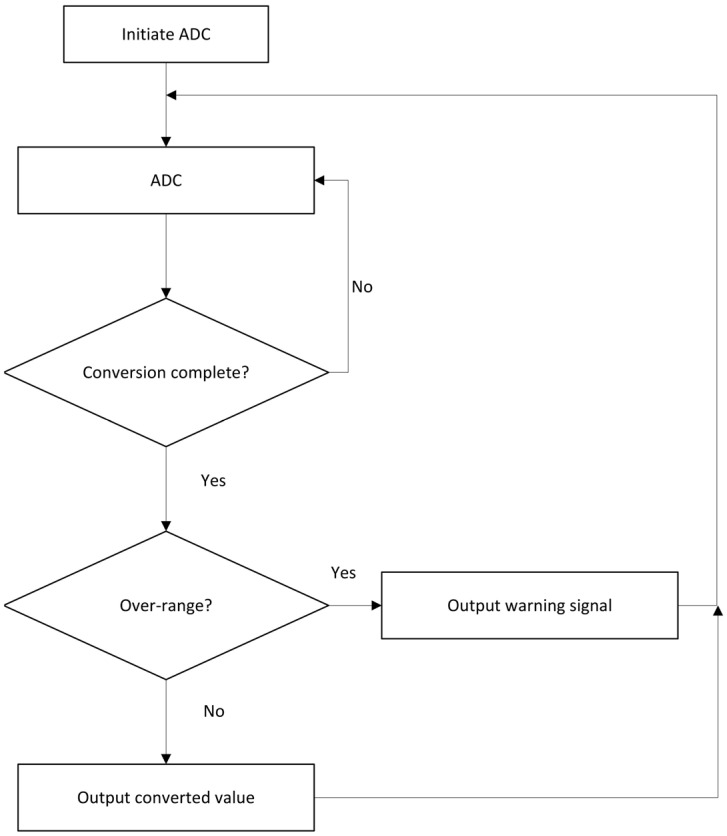
Sampling flowchart of ADC.

**Figure 9 sensors-25-07458-f009:**
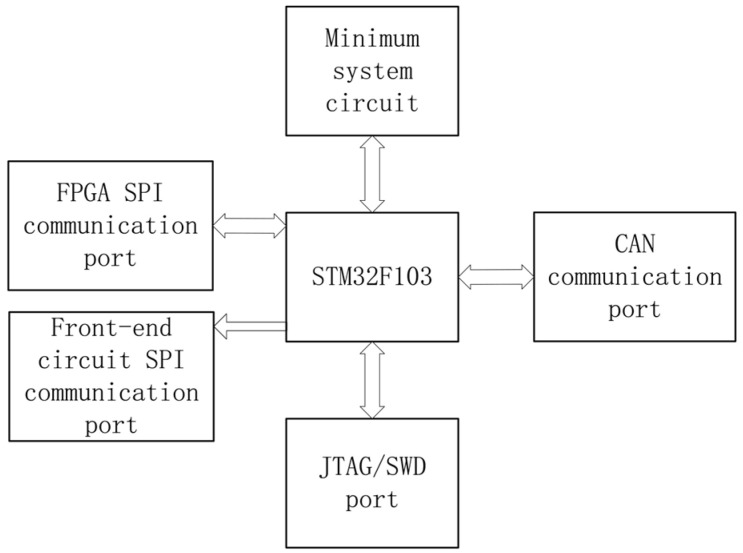
Block Diagram of MCU Peripheral Circuits.

**Figure 10 sensors-25-07458-f010:**
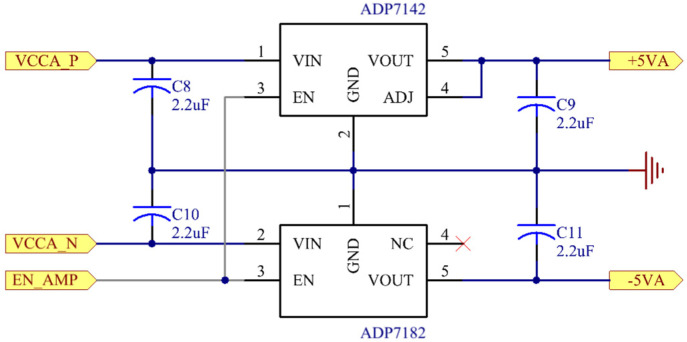
Low-voltage analog power supply circuit.

**Figure 11 sensors-25-07458-f011:**
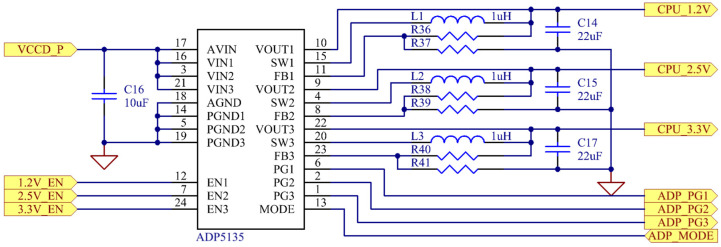
Low-voltage digital power supply circuit.

**Figure 12 sensors-25-07458-f012:**
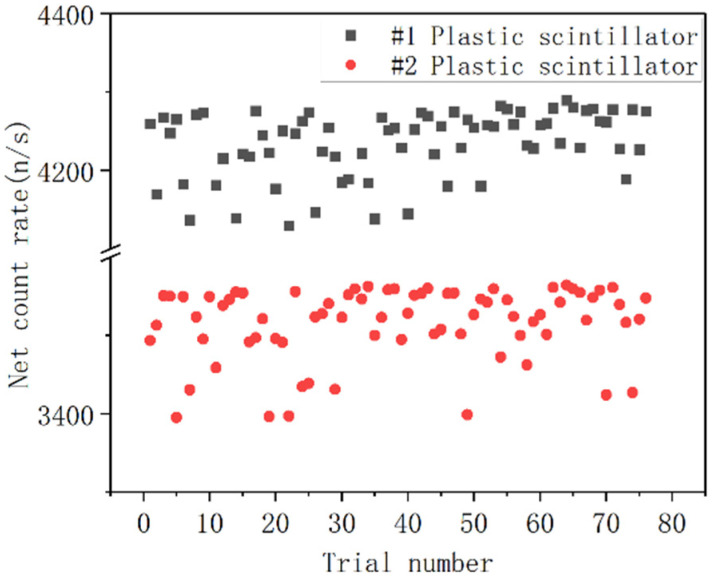
Background Counting with Two Sets of Plastic Scintillators.

**Figure 13 sensors-25-07458-f013:**
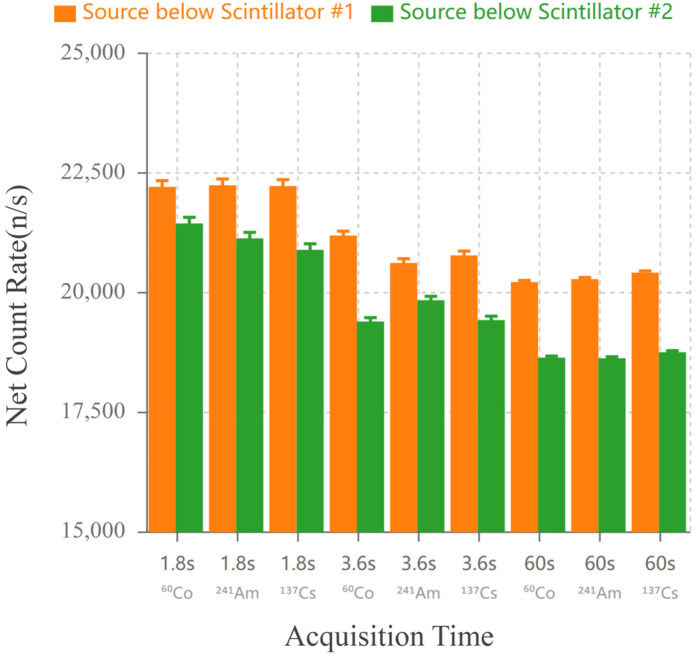
Net count rate comparison for three radioactive sources (^60^Co, ^241^Am, ^137^Cs) measured at different acquisition times (1.8 s, 3.6 s, 60 s) after background subtraction. Orange bars: source positioned below Plastic Scintillator #1; Green bars: source positioned below Plastic Scintillator #2.

**Figure 14 sensors-25-07458-f014:**
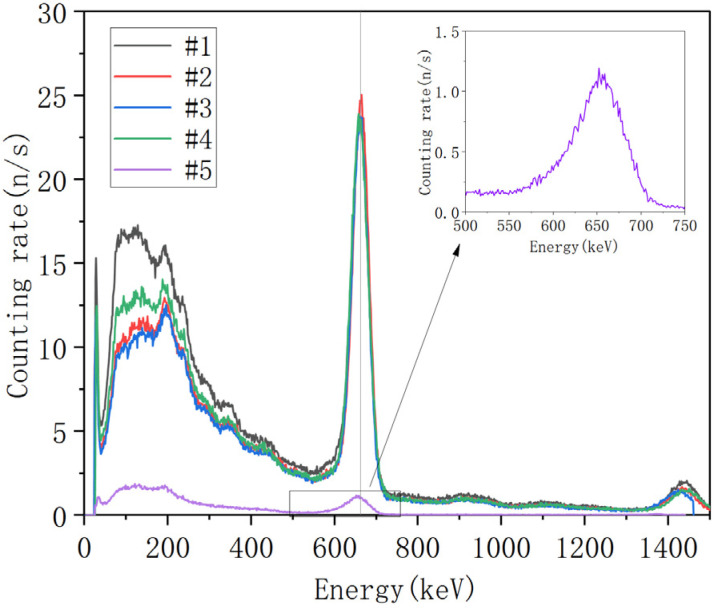
Energy Spectrum of the ^137^Cs Source.

**Figure 15 sensors-25-07458-f015:**
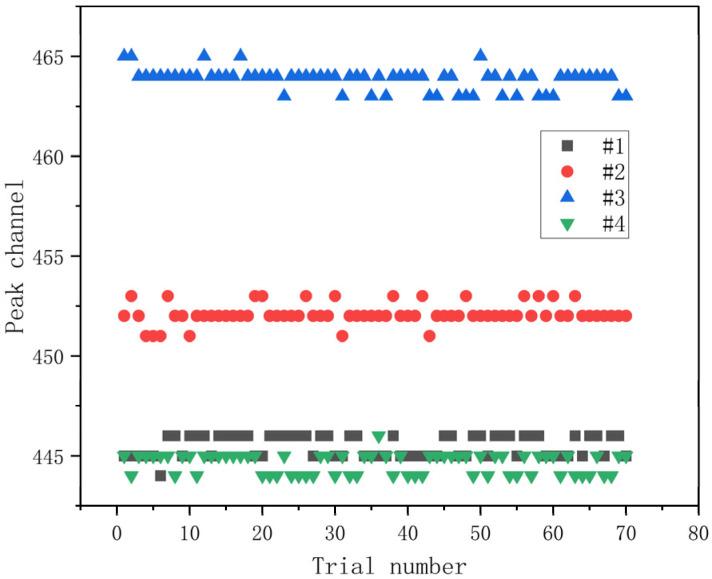
For each repeated measurement, the channel address corresponding to the ^137^Cs peak detected by the NaI array detector.

**Figure 16 sensors-25-07458-f016:**
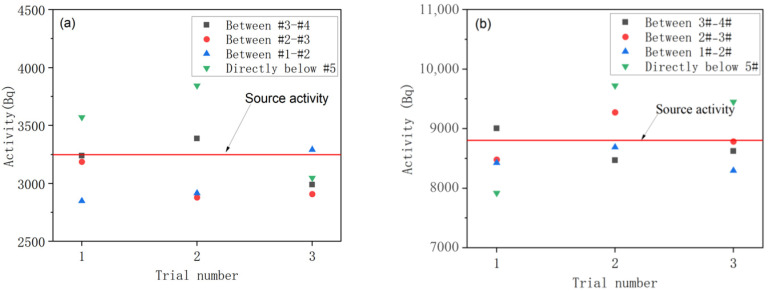
Activity measurements for (**a**) ^241^Am and (**b**) ^137^Cs.

**Table 1 sensors-25-07458-t001:** Information on radioactive sources used in experiments.

Nuclide	Serial Number	Activity (Bq)	Specifications	Calibration Date
^60^Co	1948056	8.36 × 10^4^	Φ 32 × 3 mm	1 July 2019
^241^Am	2143502	3.25 × 10^3^ *	Φ 15 mm	25 January 2021
^137^Cs	2143501	8.80 × 10^3^	Φ 15 × 3 mm	25 January 2021

* Converted from emission rate of 1.95 × 10^5^/2π·min.

**Table 2 sensors-25-07458-t002:** Energy Calibration Parameters for 5 NaI Detectors.

Detector	Location of Characteristic Peaks		Calibration Coefficients	
	(59 keV)	(1.173 MeV)	*a*	*b*
#1 Nal	38	785	1.479	2.541
#2 Nal	39	788	1.467	1.205
#3 Nal	40	811	1.425	1.415
#4 Nal	37	780	1.479	3.735
#5 Nal	41	864	1.403	3.713

## Data Availability

The measurement data generated during the experiments were used to validate the proposed radioactive sorting and volume reduction system. Due to the project’s confidentiality requirements, the raw data cannot be shared publicly. Summarized experimental results relevant to system performance are fully presented in this article.

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
