# Peer review of "Development of a Radioactive Sorting and Volume Reduction System for Radioactive Contaminated Sandy Soil Using Plastic Scintillator and NaI Detectors"

_sensors, 2025, doi:10.3390/s25247458_

Round 1

Reviewer 1 Report

Comments and Suggestions for Authors

This manuscript proposes a radioactive measurement and sorting system for radioactive sandy soil, employing a differential measurement scheme with dual plastic scintillators to achieve dynamic background subtraction. This innovative solution enables rapid radioactive identification within short measurement cycles. The system design is comprehensive and meticulous, incorporating a signal conditioning circuit with programmable gain control and temperature stabilization capabilities, fully reflecting engineering considerations for field deployment conditions. Overall, this study provides valuable technical contributions to the development of radioactive sand-soil separation equipment, holding significant practical significance. However, several aspects require clarification and additional analysis to strengthen the scientific rigor and practical applicability of the findings.

  1. The manuscript mentions the temperature stabilization system multiple times, but it appears to be shown only in Figures 2 and Figure 3 without any corresponding textual description. It is recommended that the authors provide additional explanation.
  2. Section 4 briefly mentions the SGS system but lacks systematic comparison. Please provide a comparison table summarizing key performance metrics: detection limits, processing speed, sorting precision, applicable radionuclides, cost considerations, etc. This would help readers better understand the advantages and limitations of your approach relative to the state-of-the-art.
  3. Lines 348-359 discuss the occurrence of negative net count rates, but the explanation contains logical errors: The text states "when Plastic Scintillator #2's reading, after correction by coefficient k, marginally exceeds Plastic Scintillator #1's reading" - this description is confusing and appears backwards. Suggest revising this paragraph.
Comments on the Quality of English Language

The English could be improved to more clearly express the research.

Author Response

Dear Reviewer,

Thank you very much for taking the time to review this manuscript. We really appreciate your excellent suggestions to improve our manuscript. All the suggestions and comments have been responded point-by-point during the revision. Please see the attachment.

Reviewer 2 Report

Comments and Suggestions for Authors

This paper focuses on development and testing of detectors for sorting of contaminated soil.

Was the plastic scintillator PVT or polystyrene?

What shielding was used above and below the detectors to reduce background?

How were the light signals from the scintillators converted to electrical signals? Phototubes? SiPMs? What model and characteristics?

How is the thickness of the material measured on the belt controlled? This directly impacts the measurement result. Is the measured material volume controlled or is a contimuous stream of material measured?

Paragraph at line 237: How far were the sources from the detectors? Why not farther away to produce a signal from the entire detector?

Figure 12: what are the statistical uncertainties for each measurement?

Section 3.3: how were the activities determined by the detectors computed? Was a simulation performed to make this conversion? Or was some formula used?

Paragraph at line 362: How was temperature used to adjust the gain? Equation used? Experimentally determined?

Author Response

Dear Reviewer,

Thank you very much for taking the time to review this manuscript. We really appreciate your excellent suggestions to improve our manuscript. All the suggestions and comments have been responded point-by-point during the revision.

All modifications have been highlighted in red in the revised manuscript. Key responses that are also incorporated into the main text are presented in italics in this document for the reviewers' convenience. In addition to addressing the reviewers' comments, other modifications we made to improve the manuscript are also highlighted in red. Our responses are as follows.

Round 2

Reviewer 1 Report

Comments and Suggestions for Authors

N/A

Comments on the Quality of English Language

N/A

Reviewer 2 Report

Comments and Suggestions for Authors

Thanks for updating the paper in response to my comments.